# Beyond Pain Relief: A Cross-Sectional Study on NSAID Prescribing, Polypharmacy, and Drug Interaction Risks in Community Pharmacies

**DOI:** 10.3390/healthcare13243264

**Published:** 2025-12-12

**Authors:** Javedh Shareef, Sathvik Belagodu Sridhar, Saeed Humaid Al Naqbi, Adyan Iftekhar Bakshi

**Affiliations:** Clinical Pharmacy & Pharmacology, Ras Al Khaimah College of Pharmacy, Ras Al Khaimah Medical and Health Sciences University, Ras Al Khaimah 11172, United Arab Emirates

**Keywords:** cross-sectional study, poly pharmacy, drug interactions, non-steroidal anti-inflammatory drugs, prescriptions

## Abstract

**Highlights:**

**What are the main findings?**
Celecoxib and ketoprofen were the most commonly prescribed NSAIDs in community pharmacies, while polypharmacy and potential drug–drug interactions (pDDIs) were highly prevalent.A significant association (*p* < 0.05) was found between ibuprofen, diclofenac, aspirin, meloxicam use, and the lack of gastroprotective co-prescription, indicating a need for improved adherence to safety guidelines.

**What are the implications of the main findings?**
Reinforcing regulatory frameworks and fortifying medication-use policies, combined with pharmacist-led medication review and continuous monitoring, can mitigate NSAID-related risks.The findings provide evidence to inform rational prescribing strategies, strengthen community pharmacy practice, and enhance patient medication safety in the Ras Al Khaimah, one of the Northern Emirates of the United Arab Emirates and similar healthcare settings.

**Abstract:**

**Background/Objectives**: Non-steroidal anti-inflammatory drugs (NSAIDs) are widely used globally to manage pain and inflammation. The rising prevalence of polypharmacy and potential drug–drug interactions (pDDIs) magnified by the prolonged and irrational use of NSAIDs may jeopardize patient medication safety. This study aims to analyze the pattern in prescribing NSAIDs and assess the extent of polypharmacy and pDDIs in community pharmacies located in Ras Al Khaimah. **Methods**: A quantitative cross-sectional study was conducted in randomly selected community pharmacies over six months (July 2024 to December 2024). Prescriptions pertaining to NSAIDs were assessed for prescribing patterns; incidence of polypharmacy and pDDIs were identified using Lexicomp’s drug interaction database. Chi-square tests assessed associations between treatment variables and polypharmacy, while logistic regression explored predictors of pDDIs. **Results**: In a total of 600 prescriptions, 1865 drugs were prescribed, including 908 NSAIDs. Celecoxib (28.2%) and ketoprofen (27.6%) remained the most predominant oral and topical NSAIDs prescribed. Aspirin and celecoxib were most commonly linked with pDDIs. A total of 357 pDDIs were identified, averaging 1.87 ± 1.39 per prescription. Most were of minor severity (60.22%), risk category C (43.97%), and fair reliability (59.38%). Gender, nationality, and comorbidities were significantly associated with polypharmacy (*p* < 0.001). Logistic regression showed nationality (*p* = 0.016), comorbidities (*p* < 0.001), and drug count (*p* = 0.007) as key predictors of pDDIs. **Conclusions**: Frequent NSAIDs prescribing, incidence of polypharmacy, and pDDIs underscore the attention for more cautious, evidence-based prescribing practice. Enforcing a robust regulatory framework, coupled with strengthening medication-use policies and pharmacist-led thorough medication history review and ongoing monitoring is paramount to improve patient safety and clinical outcomes.

## 1. Introduction

Globally, individuals use non-steroidal anti-inflammatory drugs (NSAIDs) widely because of their analgesic and anti-inflammatory properties. These medications are prescribed for various conditions, such as fever and different types of pain and inflammatory disorders [1]. The availability as both prescriptions and over the counter medications and their affordability and effectiveness in various indications have made NSAIDs as integral component of modern pharmacotherapy. Despite their proven therapeutic benefits, serious adverse effects and hospital-related admissions have been reported associated with NSAIDs [2]. Their availability in multiple formations together combined with their frequent use as self-medication also contributes to incorrect use of these products, mainly in individuals with chronic pain and comorbidities. Therefore, finding the right balance between efficacy and safety is still a major problem in both clinical and community settings as patient medication safety may be jeopardized by the inappropriate or irrational use of these agents [3].

Two principal forms of NSAIDs are identified: nonselective COX inhibitors (like ibuprofen, aspirin, and diclofenac sodium/potassium) and selective COX-2 inhibitors (like celecoxib). The recent move toward selective COX-2 inhibitors is an effort to cut down on GI side effects, but these drugs still need to be prescribed with care because they can cause heart problems [4]. For people who are likely to have GI problems, clinical guidelines suggest using selective COX-2 inhibitors or NSAIDs with drugs that protect the stomach. For people with substantial GI problems, doctors generally recommend using both COX-2 inhibitors and proton pump inhibitors (PPIs). Even though these are relevant approaches, many people, especially those with heart problems, overuse NSAIDs and do not utilize PPIs correctly [2,5].

Various studies carried out globally demonstrated the clinical risks associated with prescribing NSAIDs inappropriately. Co-prescribing of NSAIDs with other medications such as oral anticoagulants, angiotensin-converting enzyme (ACE) inhibitors, angiotensin receptor blockers (ARBs), diuretics and steroidal preparations can lead to potential drug–drug interactions (pDDIs) and escalate the risk of adverse outcomes affecting gastrointestinal, cardiovascular and renal systems [1,3,4]. The drug-related problems (DRPs) associated with NSAIDs reported in Canadian and Iranian studies further underlines the necessity for cautious prescribing, pharmacist-led monitoring and healthcare education regarding NSAID usage [6,7].

The rising prevalence of polypharmacy, defined as the concurrent use of five or more medications, is compounded by the prolonged and potentially inappropriate use of NSAIDs [8]. Polypharmacy increases the risk of DRPs which may, in turn, lead to higher hospitalization rates and greater healthcare expenditure in particularly elderly demographics and individuals with multiple comorbidities [9]. Therefore, analyzing prescribing patterns and identifying the prevalence of polypharmacy and pDDIs are vital components of clinical practice to optimize drug therapy and enhance therapeutic outcomes.

Irrational or inappropriate prescribing of non-steroidal anti-inflammatory drugs (NSAIDs) remains a global concern, particularly in community pharmacy practice. In the United Arab Emirates (UAE), where a diverse expatriate population frequently relies on pharmacies for the management of acute and chronic pain, pharmacists play a crucial role in ensuring the safe and rational use of these medicines. Because NSAIDs are often co-prescribed with medications for chronic conditions such as hypertension, diabetes, and gastrointestinal disorders, patients are at increased risk of potential drug–drug interactions (pDDIs). Although a regulatory framework exists, variable enforcement and the common perception that NSAIDs are inherently safe may lead to misuse and under-recognition of interaction-related risks. However, comprehensive local data describing NSAID prescribing patterns and their association with pDDIs in community pharmacy settings remain scarce [10].

Systematic research analyzing the NSAIDs that are prescribed, the occurrence of polypharmacy, and the frequency of pDDIs may help to bridge the gaps and to improve patient safety, guide policy, and update evidence-based interventions. Furthermore, it may help healthcare professionals in formulating educational interventions, thereby improving adherence to therapeutic guidelines and promoting use of NSAID rationally. Community pharmacies refer to private retail pharmacies that provide outpatient dispensing, medication counseling, and over-the-counter services to the general public. These settings typically serve ambulatory patients and operate independently of hospital-based or institutional healthcare facilities.

Therefore, this study aims to analyze the utilization of NSAIDs in private community pharmacies located in Ras Al Khaimah, one of the Northern Emirates of the United Arab Emirates. It focuses on identifying commonly prescribed NSAIDs based on type, dosage, route, frequency, duration, and their use as monotherapy or in combination. The study also examines pDDIs, patient clinical characteristics, and the prevalence of polypharmacy and its correlated risk factors. Overall, the findings aim to inform safer, more rational, and cost-effective NSAID use in community pharmacy settings.

## 2. Materials and Methods

### 2.1. Study Design and Setting

A cross-sectional, non-interventional research was performed over six months (July 2024 to December 2024) in various private community pharmacies located in Ras Al Khaimah, one of the Northern Emirates of the United Arab Emirates.

### 2.2. Sample Size

Epi Info™ software version 7.2, made by the Centers for Disease Control and Prevention (CDC), was used to determine the sample size. We figured out the sample size by looking at how many patients with NSAID prescriptions went to private community pharmacies during the study period. The finite population correction factor was based on an assumed population size of 10,000. The anticipated prevalence of the outcome factor in the population was estimated at 50%, with a confidence interval margin of ±5%. The confidence limitations were set at 5% of the total percentage points. The cluster survey has a design impact of 1. We determined that the study needed 579 participants at a 99% confidence level.

The WHO states that 600 interactions are the bare minimum needed for prescription analysis [11,12]. Thus, 30 prescriptions were chosen at random from each pharmacy to analyze the 600 prescriptions in the current study.

### 2.3. Sampling Technique

A cluster random sampling approach was employed to select community pharmacies across five geographical regions of Ras Al Khaimah, with six pharmacies randomly chosen from each region to ensure representative coverage. Within each selected pharmacy, systematic random sampling was then applied to collect eligible prescriptions. Every nth prescription (based on daily dispensing volume) that contained at least one NSAID was included until the required sample size was achieved. This multistage approach minimized selection bias and enhanced the representativeness of the data across the study area.

From each selected community pharmacy, approximately 40 prescriptions containing at least one NSAID were systematically sampled during the study period. Each prescription represented a unique dispensing encounter, and only one prescription per patient was included to avoid duplication and overrepresentation. When multiple prescriptions from the same patient were encountered, only the first eligible prescription was considered for analysis. This approach ensured independent observations and minimized sampling bias.

### 2.4. Criteria for Inclusion and Exclusion

This study included all prescriptions for adult patients aged ≥18 years who had been given at least one NSAID (systemic or topical); these were included to provide a comprehensive overview of NSAID prescribing patterns and potential drug–drug interactions. All the people that took part gave their informed consent.

Prescriptions not containing any NSAIDs, prescriptions containing NSAIDs issued from hospitals or primary eHealth centers and NSAIDs prescriptions with insufficient or incomplete patient identification were not considered for this study.

### 2.5. Study Procedure

A data collection form was used to document socio-demographic, disease, and treatment-related endpoints.

#### 2.5.1. Assessment of Prescribing Pattern

Our study region has about 150 private community pharmacies, which are spread out over five main geographical areas of Ras Al Khaimah. For the study, at least five pharmacies from each location were chosen at random. This was based on how many pharmacies were available and how keen community pharmacists were to take part. After this stratified random sample, the study included 30 pharmacies. We gathered prescriptions made by a variety of medical professionals, such as general practitioners and specialists. The information on the prescription was moved to the form that was made for collecting data. Also, the prescriptions that were filled had all the information that was needed, such as demographic and clinical data including diagnosis, comorbidities, and pharmacological therapy.

We looked at data about the pharmaceuticals prescribed (class and individual NSAIDs, dosage form, length of therapy, mono/combination therapy, etc.) and the elements in the prescription in order to meet the study’s goals.

#### 2.5.2. Assessment of Polypharmacy

Researchers assessed the number of medications patients were taking by counting the drugs dispensed at the outpatient pharmacy. Polypharmacy was categorized into three levels: ‘non-polypharmacy’ (fewer than five medications), ‘polypharmacy’ (five to nine medications), and ‘hyper-polypharmacy’ (≥10 medications). This is because there is not one single definition of polypharmacy that everyone agrees on. Some research says that polypharmacy entails taking three or more drugs; however, this limit is not as common in clinical settings as the five-drug limit [9,13].

#### 2.5.3. Assessment of pDDIs

All medications prescribed within NSAID-containing prescriptions were analyzed for potential interactions using the Lexicomp^®^ Drug Interaction Database. All drugs administered concurrently—including NSAIDs and other co-prescribed medications—were entered into the database to identify and categorize potential drug–drug interactions (pDDIs). This approach allowed assessment of both NSAID-related and other clinically relevant interactions within the same prescription context. As per this database, the pDDIs were grouped accordingly based on the ‘severity’, ‘reliability rating’ and ‘risk rating’. The pDDIs were stratified by risk rating, beginning with “Category A” (“No interaction”), “Category B” (“No action needed”), “Category C” (“Monitor therapy”), “Category D” (“Consider therapy modification”), and “Category X” (“Avoid combination”). Also graded in accordance with “Severity” (“Minor,” “Major,” and “Moderate”) and “Reliability Rating” (“Poor,” “Fair,” “Good,” and “Excellent”) as the database specifies.

#### 2.5.4. Data Analysis

Data were analyzed using IBM SPSS Statistics version 28. Demographic, clinical, and prescribing characteristics were summarized by using descriptive statistics. To assess the associations between categorical variables such as polypharmacy, comorbidities, and the presence of pDDIs, Chi-square tests and Relative Risk (RR) calculations were used. Pearson correlation analysis was applied to explore the strength and direction of relationships between continuous variables (e.g., number of prescribed drugs and number of pDDIs). Univariate and multivariate logistic regression analyses were conducted to identify independent predictors of pDDIs. Logistic regression is a good way to model the likelihood of a binary outcome, such as whether or not pDDIs are present, depending on several factors. We chose potential confounders based on their clinical importance and what we found in the literature. This included age, gender, comorbidities (including diabetes and heart disease), and the number of drugs [14,15]. We included these factors in the multivariate model to account for any influences that might have confused the results. A *p*-value below 0.05 meant that the results were statistically significant, while a *p*-value below 0.01 meant that there was a very strong link.

## 3. Results


*Attributes of the study participants related to demographics and social factors*


In total, 600 eligible NSAID prescriptions were collected from community pharmacies across the Ras Al Khaimah region of UAE. Data on NSAIDs use by gender revealed female predominance (58.67%), with average age of 38.89 ± 13.54 years. Most patients (68.17%) were aged between 26 and 50 years. The population was predominantly of Asian origin (59.17%), followed by individuals from other Middle Eastern countries (32.83%) and UAE nationals (7%). Comorbidities were common: 481 patients had 1–2 conditions, and cardiovascular disorders were the most prevalent (50.33%), followed by musculoskeletal (33.50%) and respiratory disorders (25.33%). The mean medication count per prescription was 3.10 ± 1.44. Polypharmacy (five or more drugs) was observed in 33.5% of patients. Tablets were the most common dosage form (66.67%). Cardiovascular medications (31.55%), antidiabetics (22.25%), and PPIs (17.13%) were frequently co-prescribed. These findings highlight key patient characteristics and prescribing patterns relevant to NSAID use in community pharmacy settings (Table 1).


*Clinical status of the patients under study*


Overall, NSAID prescriptions were predominantly associated with musculoskeletal and inflammatory pain conditions and were frequently issued to individuals with multiple comorbidities. Shoulder, joint, and low-back pain (45.5%), spondylosis/radiculopathy (26.0%), and arthritis-related pain (16.8%) were the most common pain indications. Among comorbidities, cardiovascular (50.3%), musculoskeletal (33.5%), and respiratory (25.3%) disorders were most prevalent. These results suggest that NSAIDs were mainly utilized for the management of musculoskeletal pain in patients with concurrent chronic diseases (Table 2).


*Usage patterns of NSAIDs medications*


Among the 600 prescriptions, 908 NSAID medications were prescribed, and on average each prescription had 1.51 ± 0.65 NSAIDs. One NSAID was the least amount prescribed to each patient, while four was the most. Out of the 600 prescriptions, most of the study participants (342, or 57.0%) received one NSAID medicine for each prescription. The next most common group (210, or 35.0%) received two NSAIDs per prescription.

The most prevalent kind of medicine prescription was non-selective COX inhibitors (465, or 51.21%), followed by selective COX inhibitors (220, or 24.22%). Celecoxib was the most common oral NSAID, while ketoprofen was the most common topical NSAID (Table 3).


*Prescribing frequencies of NSAIDs based on the age group and gender*


According to the patients’ age group, ibuprofen was most frequently prescribed in study populations under the age of 25 whereas celecoxib was most predominantly prescribed in the age groups between 26 and 75 years. Ketoprofen was the frequently prescribed topical NSAID among all the age groups (Table 4).


*Co-prescribing Gastroprotective Agents with NSAIDs*


Only 27.33% of the prescriptions contain proton pump inhibitors (PPIs—omeprazole 50.6%, dexlansoprazole 26.8%, esomeprazole 17.6%) along with NSAIDs. The remaining 72.67% of prescriptions did not have any gastro-protective drug.


*Factors Associated with Gastroprotective (PPI) Co-Prescribing among NSAID Users*


The co-prescription of proton-pump inhibitors (PPIs) with NSAIDs was significantly influenced by patient age, comorbidity burden, and the type of NSAID prescribed. Older adults (>50 years) and those with comorbid conditions were more likely to receive gastroprotective agents (*p* < 0.05). PPI use was particularly common with systemic NSAIDs such as aspirin, meloxicam, ibuprofen and diclofenac. (Table 5)


*Prevalence of pDDIs*


Analysis of gastroprotective medication use revealed that proton-pump inhibitors (PPIs) were more frequently co-prescribed to older adults (≥51 years) and to patients with multiple comorbidities, particularly cardiovascular, gastrointestinal, or metabolic disorders. Co-prescription rates were also significantly higher among prescriptions containing non-selective NSAIDs such as ibuprofen, diclofenac, and aspirin (*p* < 0.05). In contrast, topical NSAID users rarely received PPIs, reflecting their lower risk of GI complications. These findings indicate that prescribers tend to selectively provide gastroprotective therapy for higher-risk patient groups and systemic NSAID users.


*Factors Associated with Gastroprotective (PPI) Co-Prescribing*


Of the 600 prescriptions, 190 (31.66%) had potential drug–drug interactions (pDDIs), most commonly 1–2 pDDIs (150 prescriptions), while fewer had ≥3 interactions. A total of 357 pDDIs were identified involving 41 drug pairs. Of these, 1.68% were significant, 38.09% moderate, and most required monitoring (category C: 157 pDDIs). Additionally, 70 pDDIs needed therapy modification (category D), and one was contraindicated (category X). No interactions were classified as category A (no interaction). Reliability ratings showed most pDDIs were fair (212) or good (95), with fewer rated excellent (36) or poor (14). These results show how common and important pDDIs are in the study group. (Figure 1A–C)


*Nature of pDDIs in study subjects.*


The study found 41 pairings of pharmaceuticals that interacted with one other and were linked to prescribed therapies in the patients. The most prevalent pDDIs were aspirin + celecoxib (27, 7.56%), followed by aspirin + metformin (21, 5.88%), and diclofenac topical + valsartan (18, 5.04%). (Table 6)


*Influence of demographic and clinical characteristics on the likelihood of pDDIs*


Data showed a significant association (*p* <0.05) between gender (X^2^ = 7.598; *p* = 0.007), nationality (X^2^ = 29.345; *p* < 0.001), having other health problems (X^2^ = 23.058; *p* < 0.001), the number of medicines (X^2^ = 11.642; *p* = 0.001), and the prevalence of pDDIs (Table 7).


*Relationship between the number of pDDIs and therapeutic variables*


There is a considerable positive link between the number of pDDIs and ongoing treatment-related characteristics such as gender (r = 0.113; *p* = 0.006), nationality (r = 0.151; *p* < 0.001), number of comorbidities (r = 0.203; *p* < 0.001), and the overall count of medicines received (r = 0.139; *p* = 0.001).


*Binary Logistic Regression Analysis for pDDIs*


Binary logistic regression was performed to predict the occurrence of pDDIs from all the treatment-related variables. It revealed that variables such as nationality, number of comorbidity burden and medications prescribed statistically significantly predicted pDDIs. Patients from other Middle Eastern countries had twice the odds of pDDIs compared with Asians (OR = 2.06, 95% CI 1.42–2.99, *p* < 0.001). Each additional comorbidity (OR = 1.88, 95% CI 1.31–2.68, *p* = 0.001) and each extra prescribed drug (OR = 1.57, 95% CI 1.10–2.22, *p* = 0.012) significantly increased the odds of pDDIs. Gender and age were not statistically significant (*p* > 0.05 (Table 8)).


*Link between demographic, clinical, and treatment-related factors and polypharmacy presence*


A chi-square analysis evaluated the connection between demographic, illness, and treatment-related factors with polypharmacy among the study population. Treatment-related variables such as gender (χ^2^ = 14.824, *p* < 0.001), nationality (χ^2^ = 14.087, *p* = 0.002), and number of comorbidities (χ^2^ = 8.704, *p* = 0.032) showed significant associations with the presence of polypharmacy (Table 9).


*Relationship Between Polypharmacy and pDDIs by Severity*


The results of the chi-square test demonstrate a significant correlation between polypharmacy and moderate pDDIs (*p* = 0.024), suggesting that individuals with polypharmacy are more likely to have moderate interactions. These underscore the need for regular review of prescription to mitigate the risk of polypharmacy causing moderate pDDIs. However, no significance was observed between polypharmacy and pDDIs with respect to minor and major level of severity (*p* > 0.05) (Table 10).

## 4. Discussion

Medicines remain an integral part of disease management and patient care. Minimizing the drug-related issues and enhancing the patient therapeutic outcomes can be achieved through promoting rational drug use [7]. NSAIDs rank among the most commonly used medications and require careful administration, especially in older adults and high-risk populations due to potential gastrointestinal and cardiovascular side effects [10]. A gender-stratified analysis of our research data revealed higher proportion of female patients (58.67%) experiencing pain and were more frequently prescribed NSAIDs, aligning with patterns observed in similar studies [16,17,18]. Irrespective of the age groups, NSAIDs were prescribed across all the populations with the highest frequency among adults aged 26–50, followed by older individuals, indicating a greater prevalence of use within these groups. Comparable patterns were observed in previous studies with adults aged 19–44 years showing a higher prevalence of receiving NSAIDs [19]. However, limited studies observed higher prescription rates in elderly patients, which contrasts with the current findings [20].

A higher number of NSAID recipients had comorbidities, primarily cardiovascular (CV) and musculoskeletal disorders aligning with the findings from preceding studies [17,21,22]. Hypertension was predominant among patients with CV and prescribing NSAIDs may further elevate blood pressure, necessitating the need for continual monitoring during prolonged therapy for patient safety. Evidence shows that even short duration, patients who have had a heart attack and use NSAIDs are 1.45 times more likely to have another heart attack or die, especially during the early treatment phase as observed in a retrospective analysis [23]. In the present study, pre-existing kidney disease was identified in less than 10% of populations, highlighting the importance of monitoring renal parameters regularly and timely NSAID discontinuation to prevent further impairment.

More than half of the study population were of Asian origin, consistent with UAE demographics that include substantial expatriate communities from countries such as Bangladesh, India, Pakistan, and the Philippines. Many individuals from these communities are employed in sectors like construction, healthcare, and hospitality, and frequently access community pharmacies. Typically, independent community pharmacies remain the first entry point of contact for various health concerns for these individuals, particularly pain, swelling and various inflammatory disorders. Cultural and linguistic familiarity with pharmacists may further influence their care-seeking behaviors and preference for accessible, over-the-counter solutions. The most prevalent reasons for receiving an NSAID prescription were pain related to arthritis and general body pain; results concur with the previous literature [24,25].


*Prescribing Patterns of NSAIDs*


The average number of medications prescribed during the visit was relatively high, exceeding figures reported in earlier studies [26,27,28]. This likely reflects the complex clinical profiles of patients attending private community pharmacies. The increase in number of comorbidities often necessitates multiple medications, resulting in heightened chances of interactions and elevated medical expenses [13].

All the study populations were prescribed NSAIDs, with an average of 1.51 ± 0.65 per prescription. More than two-thirds involved multiple NSAIDs, most commonly a combination of oral and topical preparations. It should be noted that prescribing multiple NSAIDs offers no additional therapeutic benefit; rather, it may elevate the risk of adverse outcomes, as documented in previous studies [21,25,29].

Ibuprofen and ketoprofen remained the most frequently prescribed oral and topically NSAIDs, ranking highest in prescription frequency. The quick symptom-relieving effects combined with lower systemic risk could be the reason for the preference of these topical formulations. This prescribing pattern corroborates findings from prior studies [30,31]. In contrast, a study from Yemen reported diclofenac had the highest NSAID prescription rate whereas naproxen predominated in Nepal [32,33]. The broader dataset used in the present study, derived from multiple community pharmacies, provides a more comprehensive perspective as compared to those studies that are limited to orthopedic outpatient departments. The favorable gastrointestinal safety profile of celecoxib, a selective COX-2 inhibitor, could be the reason for its most frequent prescription and wide use over traditional NSAIDs [4].

A statically significant pattern was observed in our study with adults receiving higher prescriptions of celecoxib, ibuprofen, and ketoprofen, while lower prescription rates were noted among the aging populations. This shows that prescribers exercised caution due to growing safety concerns about NSAID use in elderly demographics at risk of developing cardiovascular, renal, and gastrointestinal complications [17,25]. To limit the chance of gastrointestinal complications, standard NSAIDs ought to be co-prescribed with PPIs or substituted with selective COX-2 inhibitors. Prescribers are advised to evaluate risk factors such as age over 45; concurrent use of anticoagulants, aspirin, or corticosteroids; and a history of gastrointestinal or cardiovascular events [8,10]. In our study, only 27.33% of prescriptions received a PPI alongside NSAIDs. Similar low co-prescription rates have been reported in previous studies (36% and 40%) [34,35]. In contrast, another study reported a significantly higher co-prescription rate of 72% [36].

Among prescriptions without PPI co-prescription, 34.40% were given selective COX-2 inhibitors, and 38.26% were taking traditional NSAIDs, putting them at increased risk of GI problems. Statistical analysis revealed a significant link (*p* < 0.05) between drugs like ibuprofen, diclofenac, aspirin, and meloxicam and the lack of gastroprotective co-prescription.


*Drug–Drug Interactions*


Effective and comprehensive management of chronic conditions often requires a broad range of medications for optimal symptom control. Many times, this escalates the risk of pDDIs that may lead to patient harm and adverse events and compromise therapeutic outcomes. Analyzing the patient drug therapy and identifying patient risk profiles are essential for minimizing these risks and enhancing the quality of care. The literature reports that DDIs increase rates of hospital admissions and emergency department visits [37,38]. Aligning with previous studies reporting a range of 37.6% to 66.7%, our research found a pDDIs prevalence of 31.66% in community pharmacy prescriptions [39,40]. The disparity in pDDIs prevalence between studies may be related to differences in study design, sample size, population demographics, prescribing practices, and the criteria or databases utilized to identify drug interactions. Reporting rates can also be affected by variations in healthcare settings, including hospitals versus community pharmacies, and by how prescriptions are written in different parts of the country.

As observed in our study, a higher proportion of pDDIs were minor in level of severity with less than 2% reported as significant. In contrast to this finding, a preceding study involving hospitalized patients reported three-fourths of pDDIs were moderate followed by mild in level of severity [41].

Regarding DDI risk assessment, most interactions in this study were Category C, indicating the need to monitor drug therapy. Similar patterns have been reported in previous studies [42,43]. In contrast, another study reported most pDDIs as Category D, where modification of therapy is recommended, reflecting a higher level of clinical concern due to increased severity [44]. Furthermore, only 0.28% of prescriptions contained drug combinations categorized as Category X (“avoid the combination”), comparable to 0.36% reported in another study but lower than rates in other investigations (3.02% and 16.5%) [43,45].

DDIs occurred when NSAIDs were taken with medications that caused moderate to significant interactions, including statins, cardiovascular drugs, anti-diabetic medications, proton pump inhibitors, vitamins, and nutritional supplements, in decreasing order of prevalence. These combinations are occasionally necessary and recommended if the patient has concomitant conditions such as diabetes mellitus or cardiovascular disease. Furthermore, because both NSAIDs and antihypertensives affect the kidneys, their combination may raise the risk of renal impairment in addition to effects on blood pressure [41]. Therefore, careful patient monitoring—including observation of signs and symptoms, laboratory parameters, and ongoing evaluation of the benefits versus risks of continuing medications—is essential. Additionally, a clinical decision support system is recommended to assist prescribers in managing pharmacotherapy for pain-related disorders, ultimately enhancing patient safety [46].

The correlation between treatment-related factors and possible DDIs revealed that comorbidities, number of medicines, gender, and nationality all considerably raised the possibility of drug interactions, aligning with the previous literature [9,10,40]. An Ethiopian study found a notable link between drug interactions, the count of dispensed drugs, and the existence of comorbid conditions [47]. Similarly, our research indicates that patients prescribed five or more medications are inclined to experience pDDIs versus those taking fewer than five.

A substantial uptick in the prevalence of potential DDIs was documented with a rising number of prescribed medications and comorbidities, as shown by multivariate logistic regression analysis (*p* < 0.05). Previous studies have also reported that older adult females are more frequently prescribed NSAIDs [44]. The utilization of multiple drug therapy to manage comorbid conditions contributes to an elevated risk of pDDIs. Therefore, prescribers should be mindful of the risk factors connected to pDDIs. A multidisciplinary approach involving close and continuous monitoring can help optimize pharmacotherapy and reduce the occurrence of adverse interactions and associated side effects. This investigation highlights the value of identifying and managing potentially harmful drug interactions to enhance clinical safety and prevent negative outcomes.


*Polypharmacy*


Polypharmacy, particularly among elderly and comorbid patients, remains a growing global concern. In our study, 33.5% of prescriptions contain five or more medications, aligning with international findings [9,10,48]. This prevalence is consistent not only with global data but also with findings from neighboring Gulf countries [49,50,51,52]. Studies in Saudi Arabia and the UAE have reported polypharmacy rates ranging from 30% to 45% among older adults in outpatient and primary care settings, reflecting similar prescribing patterns across the region in Ras Al Khaimah [36,53,54,55].

Polypharmacy was more pronounced among patients aged 60 years and older, aligning with the expected increase in medication use due to age-related multimorbidity. We also observed a significant association between polypharmacy and comorbidities, female gender, and specific nationalities, suggesting possible disparities in healthcare access, health-seeking behavior, or prescribing practices. While polypharmacy is not inherently inappropriate, it demands clinical justification, regular review, and active monitoring [52]. The risks include adverse drug events (ADEs), pDDIs, and non-adherence, especially in older adults with complex therapeutic regimens. Community pharmacists are ideally placed to detect polypharmacy, educate patients, and collaborate with prescribers to manage high-risk regimens more safely [54].

Given the accessibility of medications and prevalence of self-medication practices in the Ras Al Khaimah region, a multidisciplinary approach engaging healthcare professionals, and public health authorities is essential to promote evidence-based prescribing practices. Future strategies should prioritize routine medication reviews, public education, and enhanced surveillance to mitigate risks arising from inappropriate polypharmacy. Implementing electronic prescribing systems and clinical decision support tools in community pharmacy settings could further enhance medication safety by flagging potential duplications and interactions early.

This study provides useful information about how prescriptions are written in this region by using real-world prescription data from private community pharmacies. By incorporating a wide range of people, such as men and women, different age ranges, and nationalities, it was possible to find noteworthy prescribing patterns in certain groups. One of the study’s main strengths is that it focuses on the frequency and severity of pDDIs, which is typically not mentioned enough in similar studies. The study also provides baseline data on the application of NSAIDs and the co-prescription of gastroprotective agents. This is useful for future pharmacoepidemiologic studies and policymaking in the area.

This study has some limitations. The cross-sectional design makes it hard to observe how prescribing habits change over time or to evaluate long-term patient outcomes. The investigation only looked at prescription data, excluding over-the-counter NSAID use, patient adherence, or unreported comorbidities. This may have led to underestimation of the true level of NSAID exposure and DDI risk. The study did not assess clinical outcomes such as hospitalizations or adverse drug reactions, making it harder to determine patient safety impacts. It is also hard to interpret some results because of limited information on why prescribers chose certain drugs. Insufficient specific clinical details about the reasons for prescribed drugs made it difficult to determine whether the observed polypharmacy was clinically appropriate or avoidable. Further studies incorporating patient data and clinical outcomes would clarify the appropriateness of this pharmaceutical use. Finally, as the study was conducted in private community pharmacies in Ras Al Khaimah, the results may not generalize to public healthcare institutions or populations in other Emirates or rural areas. The inclusion of topical NSAIDs, which have minimal systemic absorption, may have led to a slight overestimation of potential interaction frequency.

## 5. Conclusions

This study evaluated the prescription patterns of NSAIDs and provides baseline data for future research in similar healthcare settings to monitor patterns in drug utilization over time. The findings suggest that COX-2 inhibitors, particularly celecoxib, were the most frequently used and remain a safer alternative to traditional NSAIDs due to their favorable gastrointestinal profile. Celecoxib and aspirin were the most common drug pair that could have drug–drug interactions (DDIs). Although most of the interactions reported were mild in nature, the presence of a few serious ones indicates that prescribers need to be vigilant while prescribing. NSAIDs ought to be prescribed at the minimal efficacious dose for the briefest feasible period. Reinforcing the regulatory framework, coupled with fortifying medication-use policies and pharmacist-led thorough medication history review and ongoing monitoring, is vital in community pharmacy settings. This approach goes a long way toward optimizing drug therapy and improving patient safety and clinical outcomes.

## Figures and Tables

**Figure 1 healthcare-13-03264-f001:**
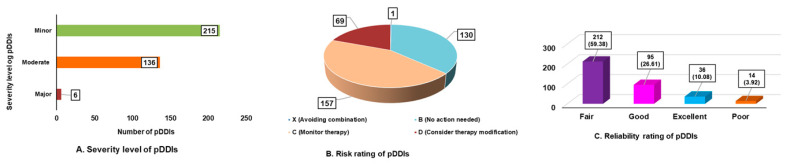
Overview of potential drug–drug interactions (pDDIs): (**A**) Severity levels of identified pDDIs; (**B**) Risk rating distribution; and (**C**) Reliability rating of pDDIs.

**Table 1 healthcare-13-03264-t001:** Socio-demographic Characteristics of the Study Population.

Variable	*n* = 600 (%)	95%Confidence Interval
**Gender**		
Female	352 (58.7)	54.7–62.6
Male	248 (41.3)	37.4–45.3
**Age (In Years)**		
≤25	84 (14.0)	11.0–16.8
26–50	409 (68.20)	64.3–72.0
51–75	99 (16.50)	13.7–19.5
>75	08 (1.30)	0.5–2.3
**Nationality**		
Asians	355 (59.2)	55.3–63.0
Arab Countries other than UAE	197 (32.8)	29.0–36.7
UAE Nationals	42 (7.0)	5.0–9.0
Others	06 (1.0)	0.2–1.8
**Number of comorbidities**		
0	26 (4.3)	2.8–6.6
1–2	481 (80.2)	76.8–83.2
3–4	80 (13.3)	10.8–15.8
≥5	13 (2.2)	1.2–3.5
**Comorbidities**		
**Diabetes mellitus**		
Yes	24 (4.0)	2.5–5.7
No	576 (96.0)	94.3–97.5
**Hypertension**		
Yes	102 (17.0)	13.8–20.0
No	498 (83.0)	80.0–86.2
**Dyslipidemia**		
Yes	176 (29.3)	25.8–33.0
No	424 (70.7)	67.0–74.2
** Thyroid disorders **		
Yes	48 (8.0)	5.8–10.2
No	552 (92.0)	89.8–94.2
** Gastrointestinal diseases **		
Yes	81(13.5)	10.8–16.5
No	519 (86.5)	83.5–89.2
** Kidney diseases **		
Yes	26 (4.3)	2.8–6.2
No	574 (95.7)	93.8–97.2
** Anemia **		
Yes	13 (2.2)	1.0–3.3
No	587 (97.8)	96.7–99
** Asthma **		
Yes	53 (8.8)	6.5–11.2
No	547 (91.2)	88.8–93.5
**Chronic Obstructive pulmonary Disease**		
Yes	5 (0.8)	0.2–1.5
No	595 (99.2)	98.5–99.8
** Orthopedic **		
Yes	184 (30.7)	26.8–34.2
No	416 (69.3)	65.8–73.2
** Neurological **		
Yes	22 (3.7)	2.3–5.3
No	578 (96.3)	94.7–97.7
** Psychiatric **		
Yes	25 (4.2)	2.7–6.0
No	575 (95.8)	94.0–97.3
** Dermatological **		
Yes	19 (3.2)	1.8–4.8
No	581 (96.8)	95.2–98.2
** Muscle spasm **		
Yes	37 (6.2)	4.2–8.2
No	563 (93.8)	91.8–95.8
**Respiratory infections**		
Yes	94 (15.7)	12.8–18.8
No	506 (84.3)	81.2–87.2
**Glaucoma**		
Yes	34 (5.7)	3.8–7.7
No	566 (94.3)	92.3–96.2

**Table 2 healthcare-13-03264-t002:** Types of Diagnosis among the Study Patients.

Sl No	Diagnosis	Frequency (*n* = 600)	%
	**Pain-related conditions**		
1	Pain in shoulder/joint/elbow/low back/body ache	273	45.5
2	Spondylosis/radiculopathy	156	26.0
3	Arthritis related pain	101	16.83
4	Dental/tooth/gingivitis	21	3.50
5	Pain unspecified disease/disorder	11	1.83
6	Chronic diseases related pain (gout, inflammatory bowel disease, diabetes mellitus, iron deficiency anemia)	14	2.33
7	Osteoporosis/hemorrhoids	08	1.33
8	Neuropathic pain	07	1.16
9	Diagnosis unspecified	09	1.50
	**Common comorbidities**		
	1. Cardiovascular diseases	302	50.3
	2. Musculoskeletal disorders	201	33.5
	3. Respiratory disorders	152	25.3
	4. Diabetes mellitus	24	4.0
	5. Hypertension	102	17.0
	6. Dyslipidaemia	176	29.3

**Table 3 healthcare-13-03264-t003:** Therapeutic Class of NSAID Medications with Individual Drugs.

Drugs Category	ATC Code	*n* = 908	%	CI 95%
**Non-selective COX inhibitors**				
Aspirin	B01AC06	75	12.5	10.0–15.3
Ibuprofen	M01AE01	90	15.0	12.2–18.3
Indomethacin	M02AA23	18	3.0	1.7–4.5
Ketoprofen	M02AA10	224	37.3	33.5–41.3
Piroxicam	M01AC01	58	9.7	7.3–12.2
**Preferential COX-2 Inhibitors**				
Diclofenac	M01AB05	126	21.0	17.5–24.3
Meloxicam	M01AC06	95	15.8	13.0–18.7
**Selective COX-2 inhibitors**				
Celecoxib	M01AH01	220	36.7	33.2–40.7

**Table 4 healthcare-13-03264-t004:** Prescribing frequencies of non-steroidal inflammatory drugs according to the age group of the patients.

NSAIDs	≤25 Years	26–50 Years	51–75 Years	>75 Years	Total (*n* = 908)	*p*-Value
Aspirin	4 (5.33)	50 (66.66)	21 (28.0)	0 (0.0)	75	0.008
Celecoxib	19 (8.63)	158 (71.81)	37 (16.81)	6 (2.72)	220	0.004
Diclofenac	19 (15.07)	88 (69.84)	17 (13.49)	02 (1.58)	126	0.769
Ibuprofen	27 (30.0)	56 (62.22)	7 (7.77)	0 (0.0)	90	<0.0001 **
Indomethacin	3 (16.66)	12 (66.66)	3 (16.66)	0 (0.0)	18	0.947 **
Ketoprofen	17 (7.58)	161(71.87)	42 (18.75)	4 (1,78)	224	0.004
Meloxicam	17 (17.89)	66 (69.47)	12 (12,63)	0 (0.0)	95	0.274
Piroxicam	4 (6.89)	31 (53.44)	21 (36.20)	2 (3.44)	58	0.001

Chi-square test; ** Fischer’s exact; *p* < 0.05 was considered significant; NSAID: Nonsteroidal anti-inflammatory drug.

**Table 5 healthcare-13-03264-t005:** Association between types of NSAIDs prescribed and whether it is co-prescribed with a gastroprotective agent (s).

Variables	Co-Prescription with Gastro-Protective Agents			*p*-Value
	Yes	No	Total (*n* = 908)	
**Age group (years)**				
≤25	12 (14.3)	72 (85.7)	84	0.021
26–50	82 (20.0)	327 (80.0)	409	
51–75	52 (31.3)	114 (68.7)	166	
>75	8 (40.0)	12 (60.0)	20	
**Comorbidities**				
Present	128 (26.6)	353 (73.4)	481	**0.033**
Absent	20 (15.9)	106 (84.1)	126	
**Type of NSAIDs**				
Ibuprofen	9 (10.0)	81 (90.0)	90	**<0.0001**
Celecoxib	70 (31.81)	150 (68.18)	220	0.071
Diclofenac	23 (18.25)	103 (81.74)	126	**0.013**
Piroxicam	20 (34.48)	38 (65.51)	58	0.216
Ketoprofen	70 (31.25)	154 (68.75)	224	0.108
Aspirin	37 (49.33)	38 (50.66)	75	**<0.001**
Indomethacin	02 (11.11)	16 (88.88)	18	0.177
Meloxicam	44 (46.31)	51(53.68)	95	**<0.0001**

Chi-square test was used for analysis and *p* ≤ 0.05 was considered significant; *n* = Number of prescriptions; NSAID: Nonsteroidal anti-inflammatory drug. Bold inidicates the value is significant (*p* < 0.05).

**Table 6 healthcare-13-03264-t006:** Frequency and Nature of pDDIs in Study Population.

Sl No.	Drug Pairs	*n* = 357 (%)	Risk Rating	Severity	Reliability Rating	Mechanism of Interaction and Effect	Action
1	Clopidogrel and Esomeprazole	01 (0.28)	X	Major	Fair	Esomeprazole may diminish the antiplatelet effect of Clopidogrel.	Avoiding concurrent use due to decreased clopidogrel effectiveness.
2	Calcium carbonate and Multivitamins	03 (0.84)	D	Major	Fair	Multivitamins/Fluoride (with ADE) may increase the serum concentration of Calcium Salts.	Avoid consuming dairy products, vitamins, or supplements with calcium salts
3	Clopidogrel and Pantoprazole	02 (0.56)	C	Major	Fair	Pantoprazole may decrease the serum concentration of clopidogrel’s active metabolite.	Use pantoprazole with clopidogrel only when GI protection is clearly needed.
4	Metformin and Perindopril	14 (3.92)	C	Moderate	Poor	ACEIs may enhance the adverse/toxic effect of Metformin.	Monitor patient response to metformin closely if using these drugs concurrently.
5	Aspirin and Celecoxib	27 (7.56)	D	Moderate	Good	Aspirin may enhance the adverse/toxic effect of NSAIDs (COX-2 Selective).	High-dose aspirin is not recommended with COX-2 inhibitors beyond cardioprotective use.
6	Aspirin (Salicylates) and Metformin	21 (5.88)	C	Moderate	Fair	Salicylates may enhance the hypoglycaemic effects of blood glucose-lowering agents.	Monitor for excessive pharmacological effects (e.g., hypoglycaemia).
7	Bisoprolol and metformin	13 (3.64)	C	Moderate	Fair	Beta-blockers (Beta1 Selective) may enhance the hypoglycaemic effect of Antidiabetic Agents.	Monitor and educate patients treated with antidiabetic agents regarding the risk of masked hypoglycaemia symptoms.
8	Celecoxib and perindopril	11 (3.08)	C	Minor	Excellent	Aspirin may enhance the adverse/toxic effect of NSAIDs.	In CHF patients, consider alternative anti-inflammatory therapy to avoid NSAID-related fluid retention and edema.
9	Diclofenac (topical) and valsartan	18 (5.04)	C	Minor	Fair	NSAIDs (Topical) may diminish the therapeutic effect of ARBs.	Monitor for reduced ARB efficacy (e.g., BP, edema, renal function) when coadministered with topical NSAIDs.
10	Ketoprofen and losartan	14 (3.92)	C	Minor	Good	ARBs may enhance the adverse/toxic effect of NSAIDs.	Monitor blood pressure and renal function when ARBs are used with NSAIDs.
11	Bisoprolol and celecoxib	14 (3.92)	C	Minor	Fair	NSAIDs may diminish the antihypertensive effect of Beta-Blockers.	Monitor blood pressure changes when NSAID therapy is started, stopped, or adjusted.
12	Amlodipine and clopidogrel	12 (3.36)	C	Minor	Fair	Calcium Channel Blockers may diminish the therapeutic effect of Clopidogrel.	Monitor clopidogrel response when used with calcium channel blockers, as clinical impact and risks vary.
13	Celecoxib and valsartan	11 (3.08)	C	Minor	Excellent	ARBs may enhance the adverse/toxic effect of NSAIDs.	Consider alternative anti-inflammatory treatments in CHF patients to prevent NSAID-related fluid retention and edema.
14	Lisinopril and celecoxib	12 (3.36)	C	Minor	Excellent	ACEIs may enhance the adverse/toxic effect of NSAIDs.	Consider alternative anti-inflammatory therapy in CHF patients to avoid NSAID-related fluid retention and edema.
15	Atorvastatin and clopidogrel	14 (3.92)	B	Minor	Good	Atorvastatin may reduce clopidogrel’s antiplatelet effect.	No action is needed beyond standard clinical care measures.
16	Amlodipine and atorvastatin	21 (5.88)	B	Minor	Fair	Amlodipine may increase atorvastatin levels, while atorvastatin may reduce amlodipine levels.	No action is required beyond standard clinical care measures.

B—No action needed; C—Monitor therapy; D—Consider therapy modification; X—Avoid combination.

**Table 7 healthcare-13-03264-t007:** Bivariate Association between Treatment-Related Variables and Presence of Potential Drug–Drug Interactions.

Variable	Total Number of Patients (*n* = 600)		Chi-Square Test	
	DDI Present (*n* = 190)	DDI Absent (*n* = 410)	X^2^	*p*-Value
**Gender** ▪ Male (*n* = 248) ▪ Female (*n* = 352)	94 (37.9%) 96 (27.3%))	154 (62.1%) 256 (72.7%)	7.598	0.007
**Age Distribution (In years)** ▪ <25 (*n* = 84) ▪ 25–50 (*n* = 409) ▪ 51–75 (*n* = 99) ▪ >75 (*n* = 08)	28 (33.3%) 137 (33.5%) 25 (25.3) 00 (0.0)	56 (66.7) 272 (66.5) 74 (74.7) 08 (100%)	6.330	**0.094**
**Nationality** ▪ Asian (*n* = 355) ▪ Middle East countries other than UAE (*n* = 197) ▪ UAE (*n* = 42) ▪ Others (*n* = 06)	84 (23.7%) 90 (45.7%) 15 (35.7%) 01 (16.7%)	271 (76.3%) 107 (54.3%) 27 (64.3%) 5 (83.3%)	29.345	<0.0001
**Total Number of Prescribed Drugs** ▪ <5 drugs (*n* = 399) ▪ >5 drugs (*n* = 201)	108 (27.1%) 82 (40.8%)	291(72.9%) 119 (59.2%)	11.642	**0.001**
**Presence of Comorbidities** ▪ Comorbidity Present ▪ No Comorbidity	182 (31.7) 08 (30.8)	392 (68.3%) 18 (69.2%)	23.058	<0.0001

*p* < 0.05 is statistically significant; *p* < 0.0001 is statistically highly significant; Fischers exact. Bold indicates the values are statistically significant.

**Table 8 healthcare-13-03264-t008:** Binary Logistic Regression Analysis of Predictors of Potential Drug–Drug Interactions.

Coefficients							
Models	Unstandardized Coefficients		Standardized Coefficients	Sig	OR (Exp(B))	95% CI	
	B	Std. Error	Beta			Lower	Upper
Constant	−3.669	0.461	63.426				
Gender	0.272	0.190	2.060	0.151	1.313	0.905	1.903
Age	0.478	0.262	3.334	0.068	1.613	0.966	2.693
Nationality	0.724	0.189	14.606	0.000	2.062	1.423	2.989
No. of Comorbidities	0.628	0.183	11.773	0.001	1.875	1.309	2.684
Total no. of drugs	0.448	0.179	6.286	0.012	1.565	1.103	2.221

Dependent Variable: Presence of pDDIs; *p* < 0.05 is statistically significant.

**Table 9 healthcare-13-03264-t009:** Association between Treatment-Related Variables and Presence of Polypharmacy.

Variables	Total Number of Patients *n* = 600		Chi-Square Test	
	Polypharmacy Present (*n* = 201)	Polypharmacy Absent (*n* = 399)	X^2^	*p*-Value
**Gender** ▪ Male ▪ Female	105 (52.23) 96 (47.76)	143 (35.83) 256 (64.16)	14.824	**<0.0001**
**Age Distribution (In years**) ▪ <25 ▪ 25–50 ▪ 51–75 ▪ >75	22 (10.94) 142 (70.64) 34 (16.91) 03 (1.49)	62 (15.53) 267 (66.91) 65 (16.29) 05 (1.25)	2.376	0.500
**Nationality** ▪ Asian ▪ Middle East countries other than UAE ▪ UAE ▪ Others	99 (49.25) 84 (41.79) 17 (8.45) 01 (0.49)	256 (64.16) 113 (28.32) 25 (6.26) 05 (1.25)	14.087	**0.002**
**Comorbidities** ▪ Present ▪ Absent	190 (94.52) 11 (5.47)	384 (96.24) 15 (3.75)	8.704	**0.032**

*p* < 0.05 is statistically significant; *p* < 0.001 is statistically highly significant. Bold (*p*-values) indicates the values are significant.

**Table 10 healthcare-13-03264-t010:** Relationship Between Polypharmacy and Potential Drug–Drug Interactions (PDDIs) by Severity.

pDDIs	Polypharmacy		Chi-Square Test	
	Present	Absent	X^2^	*p*-Value
Minor	134	66	8.855	0.107
Moderate	78	122	14.311	0.024 *
Severe	08	192	3.206	0.244

* *p* < 0.05 is statistically significant.

## Data Availability

The data supporting the findings of this study are available from the corresponding author upon reasonable request. The dataset cannot be made publicly available due to confidentiality and privacy considerations of the participants, as well as restrictions imposed by the Institutional Review and Ethics Committees.

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
