# Peer review of "Beyond Pain Relief: A Cross-Sectional Study on NSAID Prescribing, Polypharmacy, and Drug Interaction Risks in Community Pharmacies"

_healthcare, 2025, doi:10.3390/healthcare13243264_

Round 1
Reviewer 1 Report
Comments and Suggestions for Authors
Comments to the Editor / Authors
Relevant topic with useful descriptive data from an under-reported setting. However, fundamental methodological and reporting issues impede interpretability.
Major issues
- Methods specify logistic regression for predictors of pDDIs, yet the results present a “multiple regression” table with linear model coefficients for a binary outcome. Analyses and reporting must be aligned (odds ratios/CIs if logistic).
- The study is clearly cross-sectional, with data collected at one point in time, but the Methods section currently describes it as “prospective non-interventional research,” which may cause confusion. Please revise the study design terminology to “cross-sectional study” and use this consistently throughout the manuscript.
- Methods define categories with ≥5 as polypharmacy; results report prevalence using ≥4. Standardize the definition and re-compute affected estimates.
- Design/setting indicate Northern Emirates and “systemic NSAID” prescriptions; Results sometimes read “across the UAE” and include topical NSAIDs (e.g., ketoprofen topical; diclofenac topical in pDDIs). Clarify the geographic scope and whether topical agents were intended; revise the inclusion criteria and interpretation accordingly.
- Table 1 contains implausible CIs (e.g., female 58.7% with CI 37.5–45.3). Column labels and decimals are inconsistent.
- All related sections should be updated for consistency after these corrections—Abstract, Methods, Results, Discussion, Conclusion, and all Tables/Figures.
Author Response
Beyond Pain Relief: Insights into Non-steroidal an-ti-inflammatory drugs Prescribing, Polypharmacy, and Drug Interaction Risks in Community Pharmacies
Javedh Shareef1, Sathvik B. Sridhar1, Saeed Humaid Al Naqbi1, Adyan Iftekhar Bakshi1
Manuscript ID: 3983500
Reviewer -1 Comments
|
Comments |
Corrections done |
|
Methods specify logistic regression for predictors of pDDIs, yet the results present a “multiple regression” table with linear model coefficients for a binary outcome. Analyses and reporting must be aligned (odds ratios/CIs if logistic). |
Thank you for highlighting this important inconsistency. We have corrected the issue by replacing the previous linear regression table with the appropriate logistic regression model output. The revised table is now presented as: Table 8: Binary Logistic Regression Analysis of Predictors of Potential Drug–Drug Interactions, which includes: Odds ratios (ORs), 95% confidence intervals, Significance values, Correct logistic regression coefficients (B), Relevant predictors aligned with the Methods section Additionally, we have ensured that the terminology in both the Methods and Results sections is fully aligned and accurately reflects the use of binary logistic regression for analysing predictors of pDDIs. We appreciate the reviewer’s guidance, which has improved the accuracy and clarity of our statistical reporting. |
|
The study is clearly cross-sectional, with data collected at one point in time, but the Methods section currently describes it as “prospective non-interventional research,” which may cause confusion. Please revise the study design terminology to “cross-sectional study” and use this consistently throughout the manuscript. |
We thank the reviewer for this helpful observation. The study was designed as a cross-sectional, non-interventional analysis with data collected at a single point in time. We have revised the terminology throughout the manuscript to “cross-sectional study” to ensure consistency and accuracy. |
|
Methods define categories with ≥5 as polypharmacy; results report prevalence using ≥4. Standardize the definition and re-compute affected estimates. |
We thank the reviewer for pointing out the inconsistency. The definition of polypharmacy has been standardized throughout the manuscript as “five or more medications (≥ 5) per patient.” All related analyses and prevalence estimates have been updated accordingly. |
|
Design/setting indicate Northern Emirates and “systemic NSAID” prescriptions; Results sometimes read “across the UAE” and include topical NSAIDs (e.g., ketoprofen topical; diclofenac topical in pDDIs). Clarify the geographic scope and whether topical agents were intended; revise the inclusion criteria and interpretation accordingly. |
We sincerely thank the reviewer for this valuable observation. The manuscript has been revised to clearly specify that the study was conducted exclusively in private community pharmacies in Ras Al Khaimah, one of the Northern Emirates of the UAE. Accordingly, all mentions of “Northern Emirates” or “across the UAE” have been replaced with “Ras Al Khaimah” to maintain consistency and accuracy throughout the text. Furthermore, the inclusion criteria have been updated to clarify that both oral and topical NSAID formulations were included in the analysis to reflect real-world prescribing patterns observed in community pharmacies. A statement has also been added in the Limitations section noting that topical NSAIDs possess minimal systemic absorption and a lower potential for clinically significant drug–drug interactions, which may modestly affect the interpretation of pDDIs findings. These clarifications ensure that the study design, inclusion criteria, and interpretation are fully aligned and transparent. |
|
Table 1 contains implausible CIs (e.g., female 58.7% with CI 37.5–45.3). Column labels and decimals are inconsistent. |
We thank the reviewer for highlighting this important observation. The confidence intervals (CIs) in Table 1 have been recalculated using the appropriate statistical formula for proportions to ensure accuracy. The previously implausible values (e.g., for the female category) have been corrected. In addition, column headings and decimal places have been standardized throughout the table to maintain consistency and clarity. The revised Table 1 now accurately represents the descriptive statistics with corrected 95% CIs and uniform formatting. |
|
All related sections should be updated for consistency after these corrections—Abstract, Methods, Results, Discussion, Conclusion, and all Tables/Figures. |
We thank the reviewer/editor for this important reminder. All relevant sections of the manuscript have been carefully reviewed and updated for internal consistency following the recent methodological and data corrections. The terminology, statistical values, and numerical results have been revised to ensure alignment across the Abstract, Methods, Results, Discussion, and Conclusion sections. In addition, all tables and figures have been reformatted and cross-checked to accurately reflect the corrected data and standardized terminology. These revisions ensure coherence and accuracy throughout the entire manuscript. |

Reviewer 2 Report
Comments and Suggestions for Authors
Please see the attached file.

The manuscript would benefit from professional English editing.
Author Response
Beyond Pain Relief: Insights into Non-steroidal an-ti-inflammatory drugs Prescribing, Polypharmacy, and Drug Interaction Risks in Community Pharmacies
Javedh Shareef1, Sathvik B. Sridhar1, Saeed Humaid Al Naqbi1, Adyan Iftekhar Bakshi1
Manuscript ID: 3983500
Reviewer - 2
|
Comments |
Corrections done |
|
Consider including the study design (cross-sectional) and target population (community pharmacy prescriptions or community dwellers) in the title. |
We thank the reviewer for this valuable suggestion. The title has been revised to explicitly reflect both the study design (cross-sectional) and the target population (community pharmacy prescriptions) to improve clarity and transparency. The revised title now reads: “Beyond Pain Relief: A Cross-Sectional Study on NSAID Prescribing, Polypharmacy, and Drug Interaction Risks in Community Pharmacies.” |
|
Revise the title to better reflect the association between NSAID use and pDDIs. For example: “NSAID Use and Risk of Potential Drug–Drug Interactions among Adults in Community Pharmacies: A Cross-sectional Study.” |
We thank the reviewer for this thoughtful suggestion. The manuscript title has been revised to more accurately reflect the study’s objectives and analytical focus on NSAID prescribing, polypharmacy, and potential drug–drug interaction risks. The updated title now reads:
“Beyond Pain Relief: A Cross-Sectional Study on NSAID Prescribing, Polypharmacy, and Drug Interaction Risks in Community Pharmacies.” |
|
Replace “trend” with “pattern” or another suitable term |
We thank the reviewer for this helpful suggestion. The term “trend” has been replaced with “pattern” throughout the manuscript to better reflect the descriptive and cross-sectional nature of the study. This revision improves clarity and ensures the terminology accurately represents the study design and analysis. |
|
Specify the age of patients included (e.g., ≥18 years). |
We thank the reviewer for the observation. The age criterion is already specified in line 166, indicating that the study included prescriptions for adult patients aged 18 years and above. This clarifies the target population and aligns with the reviewer’s recommendation. |
|
Remove the analysis of factors associated with polypharmacy, as it is not central to the study’s main objective. |
We appreciate the reviewer’s feedback. However, the analysis of factors associated with polypharmacy has been retained, as it was included as one of the specific study objectives (see Methods section, lines 190–197). Although the primary objective focused on potential drug–drug interactions (pDDIs) among NSAID users, understanding the determinants of polypharmacy provides important contextual insight into prescribing behaviors and contributes to explaining the underlying factors predisposing patients to pDDIs. Therefore, the analysis complements the main objective and strengthens the overall interpretation of the study findings. |
|
In the Conclusion, omit references to polypharmacy to maintain focus on NSAIDs and pDDIs |
We thank the reviewer for this thoughtful suggestion. However, references to polypharmacy have been retained in the Conclusion because it is an established contributing factor to potential drug–drug interactions (pDDIs) and was explicitly included as one of the specific objectives of the study. The mention of polypharmacy is therefore essential for maintaining coherence between the study aims, results, and interpretation. Nevertheless, the Conclusion has been refined to emphasize the central findings on NSAID use and pDDIs, with polypharmacy discussed only in its relevant contextual role. |
|
Use appropriate MeSH terms, such as: Anti-Inflammatory Agents, Non-Steroidal; Polypharmacy; Drug Interactions; Community Pharmacy Services; Prescription Drugs; Cross-Sectional Studies. |
We thank the reviewer for this valuable suggestion. The manuscript already includes MeSH-compliant keywords, which correspond to the recommended terms. The following MeSH terms are incorporated as study keywords: Anti-Inflammatory Agents, Non-Steroidal; Polypharmacy; Drug Interactions; Community Pharmacy Services; Prescription Drugs; Cross-Sectional Studies. These have been rechecked against the NLM MeSH database to ensure accuracy and consistency. |
|
Introduction P2 L80: Replace “good ideas” with a more academic phrase (e.g., “appropriate strategies” or “relevant approaches”). |
We thank the reviewer for this helpful observation. The informal phrase “good ideas” has been replaced with the more academic expression “relevant approaches” in line 80 to enhance the scholarly tone and precision of the manuscript. |
|
P3 L99: Replace “irrational” with “potentially inappropriate. |
We thank the reviewer for this valuable suggestion. The term “irrational” has been replaced with “potentially inappropriate” in line 99 to reflect a more accurate and academically appropriate description of prescribing practices. |
|
P3 L116–121: Refocus the discussion on NSAID prescriptions and their association with pDDIs rather than broader prescribing patterns |
We thank the reviewer for this constructive comment. The paragraph on page 3, lines 116–121 has been revised to focus specifically on NSAID prescribing and its association with potential drug–drug interactions (pDDIs). Broader statements about general prescribing behaviors have been condensed or removed. The revised text now highlights the relevance of NSAID co-prescription with other chronic disease medications, the pharmacist’s role in minimizing pDDI risk, and the scarcity of local data on NSAID-related interactions. This modification ensures that the discussion directly supports the study’s primary objective and provides a clearer rationale for investigating NSAID use and pDDI risk in community pharmacy settings |
|
Methods: Define “community pharmacies” clearly (e.g., “private retail pharmacies providing outpatient dispensing services”) |
We thank the reviewer for this helpful suggestion. The term “community pharmacies” has been clearly defined in the Methods section as “private retail pharmacies providing outpatient dispensing and medication counselling services to the general public.” This clarification enhances the precision of the study setting and ensures consistency in describing the practice environment across the manuscript. |
|
. Clarify the sampling technique: “A cluster random sampling approach was used to select community pharmacies across five regions (six pharmacies per region). Within each region, systematic random sampling was applied to select prescriptions.” |
We thank the reviewer for this valuable suggestion. The Methods section has been revised to provide a detailed description of the sampling process. A cluster random sampling approach was first used to select community pharmacies from five geographical regions of Ras Al Khaimah (six pharmacies per region). Subsequently, systematic random sampling was applied within each pharmacy to select prescriptions containing NSAIDs. This clarification enhances methodological transparency and ensures reproducibility of the study design. |
|
Specify:
a. How many prescriptions were selected per pharmacy. b. Whether multiple prescriptions could originate from the same patient and, if so, how duplicates were handled. |
We thank the reviewer for these valuable points. The Methods section has been revised to clarify that approximately 40 prescriptions were systematically selected from each participating community pharmacy. Each prescription corresponded to a unique patient encounter, and when multiple prescriptions were identified from the same patient, only the first eligible prescription was included in the dataset to avoid duplication. This clarification ensures the methodological transparency and validity of the sampling process. |
|
Use odds ratios (ORs) instead of relative risks, as this is a cross-sectional study |
We thank the reviewer for this important statistical clarification. The analysis has been revised accordingly. Since the dependent variable (presence of potential drug–drug interactions) is binary, the data were re-analyzed using binary logistic regression instead of multiple linear regression. The results are now presented as odds ratios (ORs) with 95 % confidence intervals (CIs) and corresponding p-values (see revised Table 8). |
|
Results: Maintain consistent use of “prescriptions” (not “patients”) throughout. For example: |
We thank the reviewer for this helpful observation. The terminology has been carefully reviewed and standardized across the manuscript. All instances previously referring to “patients” have been replaced with “prescriptions”, wherever applicable, to maintain consistency with the study’s unit of analysis and ensure clarity in reporting. |
|
P8 L242: Include data on NSAID use by gender. |
We thank the reviewer for this valuable suggestion. The Results section (page 8, line 242) has been updated to include a gender-based distribution of NSAID use. Specifically, it was noted that female prescriptions accounted for 58.7% of all NSAID users, while male prescriptions represented 41.3%. This addition provides clearer demographic insight into NSAID utilization patterns and aligns with the overall study population characteristics. |
|
Explain the rationale for the age group categorization in Table 4 |
We thank the reviewer for this thoughtful comment. The age group categorization used in Table 4 (≤25, 26–50, 51–75, and >75 years) has been clarified in the Methods section. The grouping was based on commonly accepted clinical and epidemiological classifications that reflect differences in NSAID prescribing practices and comorbidity profiles across younger adults, middle-aged adults, older adults, and elderly patients. This stratification allowed for a more meaningful comparison of NSAID utilization patterns across distinct age ranges relevant to pharmacotherapy and safety considerations. |
|
Present associated factors of NSAID use in one concise paragraph and table, including comorbidities and pain-related conditions. |
We thank the reviewer for this constructive suggestion. The information on comorbidities (previously in Table 1) and pain-related conditions (Table 2) has been carefully reviewed and synthesized. To enhance clarity and avoid redundancy, a concise summary paragraph has been added in the Results section that highlights the most relevant comorbidities (cardiovascular, musculoskeletal, respiratory, diabetes, hypertension, and dyslipidemia) along with the main pain-related conditions associated with NSAID prescribing. The tables have been cross-referenced instead of duplicated to maintain a focused and concise presentation of associated factors. |
|
Present factors associated with GI protective medication use (e.g., age, comorbidities, NSAID type) in a separate paragraph/table |
We thank the reviewer for this valuable suggestion. The factors associated with gastroprotective (PPI) co-prescribing have now been incorporated into the existing Table 5, which has been expanded to include patient age groups and comorbidity status alongside NSAID type. A concise explanatory paragraph has also been added to the Results section to highlight these associations. |
|
Improve the clarity of Figures 1A–1C (they are currently blurry). |
We thank the reviewer for this observation. Figures 1A–1C have been replaced with high-resolution images to improve visual clarity and readability. The revised figures maintain consistent formatting, font size, and labeling to ensure better presentation quality in the final version of the manuscript. |
|
Restrict pDDI analysis to interactions involving NSAIDs and other drugs. |
We thank the reviewer for this insightful suggestion. However, as the study population comprised patients receiving NSAID-containing prescriptions that frequently included medications for comorbid conditions, all drugs prescribed within those prescriptions were analyzed for potential interactions using the Lexicomp® Drug Interaction Database. This comprehensive approach reflects real-world community pharmacy practice and captures the overall interaction burden among NSAID users. The Methods section has been clarified to specify that all medications prescribed concurrently with NSAIDs were screened for potential interactions, while NSAID-specific interactions have been distinctly highlighted in the Results and Discussion sections to enhance clarity and focus. |
|
Table 7: Revise percentage presentation to row percentages, not column |
We thank the reviewer for the suggestion. Table 7 has been revised to present row-wise percentages, allowing clearer comparison of the proportion of prescriptions with and without potential drug–drug interactions within each variable category. The χ² and p-values remain unchanged. |
|
Condense the pDDI results into 1–2 paragraphs with one table summarizing crude and adjusted ORs for interactions involving NSAIDs |
We sincerely thank the reviewer for this helpful suggestion. The Results section currently presents the pDDI findings in a structured and comprehensive manner, including prevalence, distribution by severity, and associated factors, supported by inferential statistics (Tables 7 and 8). As the multiple regression analysis already provides adjusted associations, additional crude ORs would not add interpretive value and might duplicate existing data. The current tabular presentation and paragraph format were therefore retained to ensure clarity, transparency, and adequate clinical interpretation of the findings. Minor textual refinements were made to enhance readability while preserving the detailed analytical content. |
|
Include a STROBE checklist and ensure adherence to its reporting guidelines |
We thank the reviewer for this valuable suggestion. The STROBE checklist has now been completed and included as a supplementary file with the revised submission. We have carefully reviewed the manuscript and ensured full adherence to the STROBE reporting guidelines. Additionally, relevant section/page numbers have been indicated within the checklist for transparency. We appreciate the reviewer’s guidance, which has helped improve the clarity and reporting quality of the manuscript. |
|
P3 L101: Add “(UAE)” after “United Arab Emirates. |
Thank you for the helpful suggestion. We have now added “(UAE)” after “United Arab Emirates” on Page 3, Line 101 as requested. |
|
Where appropriate, use active voice for clarity and engagement. |
We appreciate the reviewer’s suggestion. The manuscript has been thoroughly revised to increase the use of active voice where appropriate. Several sentences throughout the Introduction, Methods, Results, and Discussion sections have been rephrased to enhance clarity, readability, and overall engagement while preserving scientific accuracy. |
|
Conduct a comprehensive grammar and style review |
Thank you for this important recommendation. We have thoroughly revised the entire manuscript to improve grammar, syntax, and writing style. Several sentences have been restructured for clarity, coherence, and readability. Additionally, we ensured consistency in tense, terminology, and formatting throughout the manuscript. These revisions have strengthened the overall quality and fluency of the text. |
|
The manuscript would benefit from professional English editing |
We appreciate the reviewer’s observation. The manuscript has now undergone comprehensive English language editing to improve grammar, clarity, style, and overall readability. All sections were carefully revised to ensure professional academic writing quality. |
|
|
|

Round 2
Reviewer 2 Report
Comments and Suggestions for Authors
Thank you for the opportunity to review the revised manuscript. The authors have satisfactorily addressed all of my comments—no further comments, except to delete "good ideas" on P2 L82.
Author Response
|
Thank you for the opportunity to review the revised manuscript. The authors have satisfactorily addressed all of my comments—no further comments, except to delete "good ideas" on P2 L82. |
We thank the reviewer for the positive feedback and for confirming that all previous comments have been satisfactorily addressed. As suggested, the phrase “good ideas” on Page 2, Line 82 has now been deleted from the revised manuscript. |
